# *Phénoménologie de la Vérité:* The Phenomenological Roots of Hans Urs von Balthasar's Theology

**Balázs M. Mezei**

Department of Political Science, Institute of Social Sciences, Corvinus University, 1093 Budapest, Hungary; balazs.m.mezei@uni-corvinus.hu

**Abstract:** Hans Urs von Balthasar, one of the leading theologians of the twentieth century, uses the methods and results of classical phenomenology in many ways. Balthasar's repeated criticism of Husserl, Scheler, and Heidegger conceals the fact of his dependence on these authors in various ways. The present text examines the implicit and explicit phenomenological elements in Balthasar's thought. As a starting point, the title of the first French translation of one of his early books, *Phénoménologie de la vérité*, is used to outline the context of Balthasar's endeavor. In what follows, I will show the phenomenological features of some of his major writings, analyze what he himself calls supernatural phenomenology, and argue for a more consistent phenomenological methodology that Balthasar could have worked out had he carefully considered the internal development of the phenomenological movement. "Apocalyptic phenomenology" emerges as the general title of an approach that links Balthasar's methodology to that of the major phenomenological works, properly examined and extended.

**Keywords:** phenomenology; Hans Urs von Balthasar; revelation; methodology; apocalyptic thinking

## 1. Introduction: Balthasar's Place in Contemporary Thought

We tend to think that Hans Urs Balthasar's place in contemporary thought is identical to his place in theology. Indeed, Balthasar[1] appears in his publications as a professional theologian, perhaps the most influential Catholic theologian of the twentieth century. But the situation is more complicated. His early work, *Apokalypse der deutschen Seele* (Balthasar 1998, untranslated), is a history and to some extent a theoretical evaluation of German Romanticism, Idealism, and early twentieth-century literature and philosophy. Although the author's background was in literary history, philosophy, and theology, the work is primarily a form of literary criticism, discussing various poets and writers from Johann Wolfgang von Goethe to Friedrich Nietzsche to Rainer Maria Rilke and Martin Heidegger. The author's general point of view is that of a theologian, but his investigations reveal the author's original background in *belles-lettres*. A similar approach can be found in Balthasar's explicitly theological works, especially the multi-volume magnum opus *The Glory of the Lord* and its sequels, *Theo-Drama* and *Theo-Logic*. Although this grandiose work presents an overarching theological vision, Balthasar is more than a traditional theologian. He is also a historian of art, philosophy, and mysticism, making him a towering scholarly figure connected in many ways to the contemporary intellectual landscape.

However, one of the central features that the reader of Balthasar's works will notice is the complexity of content and form. While the wide range of his topics is easily explained by his systematic theological aims, the form of his writings is more difficult to decipher. Sometimes it is the literary critic, sometimes the historian of theology, sometimes the biblical scholar and religious researcher who speaks to his readers. In fact, it is perhaps "the German soul" that speaks through Balthasar, a soul that kaleidoscopically presents various configurations of human richness. Balthasar's complexity, however, is methodological in a special sense: he believes that his descriptions merge into an integral methodology through

the investigation of various aspects of history, theory, logic, and aesthetics. He repeats again and again that aspects, dimensions, phenomena, and historical traditions must be carefully distinguished in order to understand their particular nature; but he adds that, beyond distinctions, a unity of form and content can be achieved.

There is widespread ambiguity among professional theologians about the place and significance of Balthasar the theologian (Schindler 1991; Oakes and Moss 2004; Nichols 2007; O'Regan 2014). His more than fifty published works on the history and problems of theology, his lifelong association with the mystic Adrienne von Speyer, and his turbulent relationship with the Jesuit order made his personality uniquely complex. However, his major works, especially *The Glory of the Lord*, place Balthasar among the most important theologians of the last century. Balthasar's outstanding theological novelty, a novelty at once theological and philosophical, is his aesthetic approach to theology. While the principle of the beautiful had some importance in earlier theological discussions, especially in the doctrine of the transcendentals of being, Balthasar's great work explained this importance in an original way and in a comprehensive context. Moreover, Balthasar's theology shows close connections not only with contemporary Catholic thinkers of the time, but also with Protestant theologians, especially with his fellow Swiss, Karl Barth (Balthasar 1992), a connection that has been taken up and continued by contemporary Protestant theologians such as Rowan Williams. While Balthasar was sometimes jokingly called "Barthasar" among his close friends, his character as a Catholic theologian is remarkably unique. On the one hand, it is the classical neoscholastic and Thomistic training that characterizes his background; on the other hand, the French *Nouvelle théologie* influenced his outlook. Jesuit scholars such as Erich Przywara, Henry de Lubac, and Karl Rahner created a context in which his work is evaluated both positively and negatively (cf. Milbank 2005; Ford and Muers 2005).

As a Swiss citizen, French was Balthasar's second language, and his reading, translation, and interpretation of French literature and theology were crucial to his intellectual development (Brisebois 1980a, 1980b). One of his mentors was the eminent French theologian, Henry du Lubac; and the most inspiring literary figures in his life were Paul Claudel and Georges Bernanos. His reception in French theology has become so widespread that is it difficult to keep track of even the most important works related to his *œuvre* (cf. Pouliquen 2005; Gabellieri 2006; Henrici 2014; Ide 2018; Peruzzotti 2021). It is important to note that French phenomenologists from Maurice Merleau-Ponty to Jean-Luc Marion and Emmanuel Falque read and used Balthasar's *théologie phénoménologique* (Balthasar 1954, p. xxvi; cf. Roten 1991, p. 84; Marion 2012, 2021; Rivera 2022; Rivera and O'Leary 2024). This point has critical importance in the present discussion concerning the possibility of improving Balthasar's theological understanding on the basis of a theologically oriented phenomenology.

"Phenomenology" in philosophy can refer either to Hegel's system of the universal workings of the Spirit (Hegel 1977) or to the development associated with the name of Edmund Husserl and his followers. It is important to look at Balthasar's understanding of Hegel (O'Regan 2014), while it is twentieth-century phenomenology that is of particular interest to me here. In the third volume of *Apokalypse*, Balthasar presents an extensive summary of the three most important early twentieth-century phenomenologists: Husserl, Scheler, and Heidegger. In his analysis, Balthasar demonstrates not only an exhaustive knowledge of the works of the mentioned phenomenologists (his familiarity with Heidegger covers only the first period of this author), but also a sophisticated understanding of the problem, the methodology, and the various developments of phenomenology. Although Balthasar does not clarify his methodology in this work—it is visibly a descriptive analysis with some evaluation—his commitments are undeniable. While he is cautious about Husserl, his sympathy for the early Scheler is obvious; but it is Heidegger who has impressed him most.

In Balthasar's later works, the influence of the phenomenology is pervasive. We can even speak of a recurring use of the phenomenological method so much that the author

himself characterizes his own aesthetics as phenomenology (Balthasar 1982, p. 499). Even more, Balthasar defines his 1947 work (Balthasar 1947) as a "*Phänomenologie der Wahrheit*" (Balthasar 1993, p. 87). Unsurprisingly, his 1947 book was translated as *Phénoménologie de la vérité* by Robert Givord (Balthasar 1952; cf. Schindler 2004, p. 99, fn. 8). Even when the theological emphasis becomes more articulate in his works, Balthasar's terminology, methodology, and references reveal his thorough knowledge of phenomenology. In this sense, Balthasar's phenomenology can be comfortably listed among the "*hérésies husserliennes*" that Paul Ricœur identified with regard to the reception of Husserl's philosophical initiative (Ricœur 2004, p. 182).

This influence of phenomenology in Balthasar's work is eminently demonstrated by what Balthasar terms "supernatural phenomenology" (Balthasar 1954, p. xviii). It contains in a nutshell a critique of classical phenomenologists in so far as they are perceived as being satisfied with a general description of what is apparent. At the same time, "supernatural phenomenology" points to the sphere of experience where phenomenology as the *ausweisende Methode*, demonstrative method, has its proper place (Balthasar 1998, vol. III, p. 272). Following some authors (Harrison 1999; Moss 2004), I will use the expression "supernatural phenomenology" as a description of Balthasar's interpretation of phenomenology. To assess the importance of Balthasarian supernatural phenomenology, we need to clarify to some extent the general meaning of phenomenology, its relationship to theology, and the context in which Balthasar's phenomenology emerged.

## 2. Levels of Phenomenology

The phenomenological movement (Spiegelberg 1982) can be divided into three major periods: 1. The first period, the Austrian-German phase, began with the work of Franz Brentano and, through the influential work of Edmund Husserl, led to the rise of the thought of Max Scheler and Martin Heidegger. 2. The second period, the French phase, began with Husserl's *Cartesian Meditations* (first published in French in 1929, cf. Husserl 1966) and developed through the work of Jean-Paul Sartre and Maurice Merleau-Ponty into the postwar work of Emmanuel Lévinas. 3. The third phase, which brought together French and Anglo-American writers in particular, can be labeled the "theological phase". This is the most influential school today, which dominates contemporary phenomenological literature, especially through the classic works of Paul Ricœur, Michel Henry, and Jean-Luc Marion (Graves 2021; Rivera 2022; Zheng 2022). As I have argued elsewhere (Mezei 2022b), what I call apocalyptic phenomenology outlines a further step in this field of phenomenological inquiry. I note that the influence of phenomenology is more widespread today than it was earlier, although rival philosophical movements (especially Marxism and, to some extent, Anglo-American analytic thought) also have their impact on the contemporary scene (Critchley 1999; Smyth and Westerman 2022).

From a methodological point of view, the following should be recalled: In the first phase, the central methodology is meticulous description, but already in Brentano's work the focus is shifted to the direct evidence of the perceiving mind by the discovery of the importance of intentionality. In the second phase, the problem of intentionality, further elaborated, gave rise to the autonomous theory of intentionality, even to the principle of the "universal *a priori* of correlation" (Husserl 1970, p. 159), the bearer of which is necessarily the transcendental *ego*. Implicit in this understanding of intentionality is the problem of being that triggered Heidegger's ontological turn. In the third period, ontology was extended to the problem of alterity, which became the central focus of many thinkers following Lévinas. The elaboration of various forms of alterity opened up the horizon of the problem of divine being. Thus, after some initial attempts, phenomenology made its "theological" turn, becoming a kind of theological phenomenology (for critical remarks, see Janicaud and Coutine 2000). The methodological patterns have also changed from description to intentional analysis, from transcendentalism to ontology, from ontology to alterity, and from the latter to the discovery of the ultimate theological horizon that defines even the simplest phenomena in our world. Apocalyptic phenomenology follows the latter

path, centering on divine revelation and elaborating the perception of this revelation as the proleptic function of revelation here and now (Mezei 2022a). It is clear from this brief summary that, contrary to opinions that emphasize the divergences and contradictions of the various phases of phenomenology's development, I see the evolution of the problems of phenomenology as more of a synergistic process, in which certain problems have triggered the emergence of other problems and, in this way, all have contributed to what I call the culmination of the phenomenological movement.

### 3. Phenomenology in Theology

The rise of phenomenology cannot be separated from theological efforts of the nineteenth century. "Phenomenology" became the title of an autonomous philosophical work with Hegel's *Phenomenology of the Spirit* (Hegel 1977). This phenomenology is deeply metaphysical and, as a matter of course, theological. As John Niemeyer Findlay aptly formulated,

> "The Christian God is essentially redemptive, and Hegel's philosophy is essentially a philosophy of redemption, of a self-alienation that returns to self in victory. If Hegel was nothing better, he was at least a great Christian theologian" (Hegel 1977, p. xvii).

Of course, one could list various opinions about the merits and demerits of Hegel's theological philosophy. What is at issue here is not a definitive theological or philosophical evaluation of Hegel's thought, but merely the importance of a phenomenology that is both metaphysical and theological, indeed a tool for a general reform of philosophy and theology at the same time. This general and productive understanding of phenomenology cannot be set aside if we want to assess the importance of phenomenology as a method for theology (Lafont 2007, pp. 63–65).

The early entanglement of phenomenology and theology explains to some extent the emerging collaboration between phenomenologists and theologians, an alliance that, while not unproblematic, was nonetheless effective, even fruitful in many ways (Rivera and O'Leary 2024). After Hegel, however, the meaning of phenomenology as a discipline and method changed significantly. This never led to a complete separation of Hegelian phenomenology from later developments (Dupont 2014), but it did lead to a different elaboration that reinterpreted the meaning and content of phenomena and their philosophical significance. With the general change in the scientific atmosphere in the second half of the nineteenth century, phenomenology turned out to be a kind of descriptive psychology, aiming at the rigorous description of perceptions and their analysis on the basis of epistemic evidence. Franz Brentano explained that the adjective is superfluous in the expression "inner perception", since all perceptions are in some sense "inner", i.e., subject-related, and only in this way can they possess evidence (Brentano 1995, p. 137). Phenomenology as descriptive psychology became a rigorous science already in Brentano's intentions. But it was Edmund Husserl who shaped "phenomenology as a rigorous science", a science that restored the role of philosophy as a fundamental and most comprehensive science (Husserl 1965).

As a rigorous science, phenomenology seemed clearly distinct from theology. Husserl claimed that the method of phenomenology requires the exclusion of "transcendence and God" from its scope of investigation (Husserl 1983, sct. 58). Nevertheless, the same author returned from time to time to the problem of religion and God and formulated possible applications of the phenomenological method with respect to such questions (Ales Bello 2009). Many of his followers, including Max Scheler, Dietrich von Hildebrand, Adolf Reinach, Edith Stein, Gerda Walther, or even Martin Heidegger, were aware of the great potential of applying the phenomenological method to the study of religion and theology. Other scholars of the same period used phenomenology as a tool for the study of religious phenomena, such as Rudolf Otto, Gerardus van der Leeuw, Gerda Walther, or Friedrich Heiler (as the latter is relatively unknown, see Heiler 1961).

Many authors who have used the phenomenological method in religious studies and theology have referred to a rather primitive methodology of some kind of description.

More sophisticated minds realized that any description presupposes a corresponding understanding, i.e., an insight, which was part of Brentano's and Husserl's methodology. The former's "inner perception" and the latter's "categorial intuition" found a fruitful reception in authors such as Scheler, Dietrich von Hildebrand, or Josef Seifert (Mertens 2018; Seifert 2024). These two levels of methodology also became influential among theologians, although the reactions in Protestant and Catholic circles were very different. While the early phenomenological movement (especially the so-called Munich Phenomenology) showed a strong Catholic influence, later phenomenology, especially the work of Heidegger, initially inspired Protestant theologians such as Rudolf Bultmann and Paul Tillich (Bultmann 1969; Tillich 2017). In Catholic circles, the still dominant scholasticism was unfriendly to phenomenology (recognizing its Augustinian roots). With the rise of the *Nouvelle théologie* from the 1920s, which pursued a rediscovery of the Platonizing Patristic authors, new waves of influence of the phenomenological method could be perceived. Balthasar analyzed this development in the third volume of *Apokalypse*. But it was Karl Rahner who organically applied Heidegger's method in his seminal work, *Hearers of the Word. Laying the Foundation for a Philosophy of Religion* (Rahner 1969).

Erich Przywara was a decisive factor in Balthasar's theological development, although the latter, while relying on the grandiose vision of *analogia entis* (Przywara 2014), criticized his mentor on a crucial point, namely in that Przywara overemphasized the importance of dissimilarity in the analogous relationship between the Creator and the creature (Balthasar 2004c, p. 95, fn. 16; for a different emphasis see Balthasar 1989, p. 37). At the same time, Przywara was a significant witness of the influence of phenomenology in theology. In *Analogia Entis*, Przywara offers insightful analyses of the differences and connections between the phenomenological methods of Husserl, Scheler, and Heidegger, pointing out the importance of noematic phenomenology (Husserl), the phenomenology of value (Scheler), and the ontological turn of phenomenology (Heidegger). The concept of analogy aims to provide an overall framework in which phenomenology in its various directions has an important place (Przywara 2014, p. 12). However, Przywara's critical assessments of phenomenology (its alleged essentialism or Heidegger's "pantheism", cf. Przywara 2014, p. 117), especially his contrast of phenomenology with "realogy", anticipate Balthasar's similar evaluations (cf. Balthasar 1991, pp. 429–50. "Realogy" is, of course, a no-word; its correct form should simply be "ontology").

We can continue this overview with later achievements, such as the hermeneutics of Romano Guardini and Paul Ricœur, or the phenomenologies of Dietrich von Hildebrand, Josef Seifert, and Jean-Luc Marion. However, it is more important to see clearly the fundamental problem of the use of phenomenology in theology. As a method, phenomenology is at its core descriptive; its descriptions aim at the data of experience. On a higher level, phenomenology is correlative analysis of the relationship between *noesis* and *noema*, which also includes the constitutive role of the transcendental *ego* (I or subject). It seems that with this emphasis Husserl transformed the phenomenological method into a subjectivist methodology. This view is wrong, because the transcendental *ego* of the universal correlation is the ideal intersection of all possible and actual intentional centers presupposed in propositions. To deny the formal existence of such an intentional center is self-refuting, because precisely by denying it we presuppose both the general validity of our assertion and the subjective expression of the act of asserting it. At the same time, Husserl neglected the importance of the most comprehensive matrix of intentionality, namely being. Therefore, Heidegger's criticism given in his ontological turn is justified; for it is also justified to say that along with the disremembering of being the factual carrier of intentionality, *Dasein*, i.e., the existing human being, is also "passed over", neglected or forgotten ("*übersprungen*", Heidegger 1996, pp. 41, 211).

Heidegger's *Phenomenology and Theology* (Heidegger 1998, pp. 39–63) offers a disjunctive assessment of the relationship between philosophy (phenomenology) and theology. A purely theological approach is historically specific and logically axiomatic, and thus presupposes (i.e., remains uninformed about) the meaning of reality. This claim is uniquely

strong in the logical sense and has never been denied by insightful theologians: theology, properly speaking, postulates a general philosophy and, in particular, a philosophy of revelation. At the same time, the immediate basis of Christian theology is the historical fact of revelation, that is, the life, death, and resurrection of Jesus. Ironically, in response to Heidegger's claim, several theologically interested and phenomenologically attuned theories have emerged, beginning with those of Johannes B. Lotz, Bernard Welte, and Karl Rahner; and later the so-called "theological" turn of phenomenology led to the thought of Emmanuel Lévinas and Jean-Luc Marion, among others (Jung and Zaborowski 2013). Heidegger's critique thus proved to be surprisingly fruitful in the sense that it triggered a process of articulating the close relationship between phenomenology and theology. As a result, phenomenology moved to the center of theologically oriented investigations, a process that still dominates vast areas of contemporary theology and philosophy (cf. Betz 2023, pp. 38–44; Rivera and O'Leary 2024).

If phenomenology is understood as a simple description of experience, theology as a deductive science (based on the logical and historical *prius* of divine revelation) will find it difficult to make room for the phenomenological method. However, if phenomenology is understood as a method for working out a plausible typology of phenomena, theology is more comfortable with it, although it still has problems with a proper interpretation of the realm of correlation. Finally, if phenomenology is understood as the opening of the ultimate horizon of being, person, alterity, or God, theology can certainly use this approach in various ways. It seems that the more we approach the third level of phenomenology, the more fruitful it becomes for theology. And the more phenomenology understands itself at the third level, the closer it comes to various versions of phenomenological theology, versions that are stretched between the explicitly theological form of some French phenomenologists on the one hand (Marion 2021), and the reformist forms of Heideggerian ontology and its sequels on the other (Rockmore 1995).

## 4. Balthasarian Phenomenology

Balthasar thoroughly read and interpreted the main figures of classical phenomenology as the third volume of *Apokalypse* attests. This is not to say that this early work is phenomenological, or his interpretations are theoretically convincing—Balthasar is rather an avid reader of the works of others in this *Jugendschrift*. However, his obvious sympathy for Heidegger makes it plausible that he indeed—like many others at that time—came under the influence of Heidegger's version of phenomenology. This influence is obvious in his 1947 work *Wahrheit. Ein Versuch*. It was no accident that the French translator of this work chose the title *Phénoménologie de la vérité* (Balthasar 1952; cf. Schindler 2004, p. 99, fn. 8). However, the general influence of phenomenology became even more palpable in his mature work, i.e., in the volumes of *The Glory of the Lord* and its sequels. In this section, I will consider Balthasar's applications of the phenomenological method in some of these works.

### 4.1. Herrlichkeit

The concept of *Herrlichkeit*, translated as glory, offers several insights into Balthasar's use of the phenomenological method. It is important to note at the outset that "phenomenon" as the central phenomenological concept is repeatedly described by German authors as *Schein*, appearance. *Schein* also means illusion, but it has a positive religious connotation as *glory* (e.g., *Heiligschein*). When a theologically inclined author, who studied the writings of the phenomenologists in depth, encountered the term *Schein*, it was not difficult for him or her to associate this term with divine glory. However, this glory can also be understood as *Herrlichkeit* (splendor, beauty) in both the empirical and the supernatural sense—despite the double meaning of *Schein* (true and false appearance [for the latter, see *scheinheilig*]).

Indeed, *Herrlichkeit* is the ultimate beauty of God's glory and its instantiations in the creation in Balthasar's work. The relationship between divine glory and empirical beauty is

analogical: the latter necessarily refers to the former while it can never be identical with it in the strict sense. There is, however, an analogical identity in which divine glory is "always greater" *(semper maior)* than what is beautiful in earthly, empirical, emotional, or spiritual terms ("*semper major est dissimilitudo quam similitudo*", as Balthasar quotes Bonaventure, cf. Balthasar 1982, p. 293).

What connects the two aspects is the concept of *Gestalt*, form, as the subtitle of the first volume also says: Seeing the Form. "Seeing" (*Schau*) has a long history in the philosophical, theological, and mystical literature. This is also reflected in Balthasar's first work (Balthasar 1998 I, 2, p. 137 [*Weltschau*]). In particular, *Schau* is central in the classical literature of phenomenology (Moran 2018; Hopp 2018). Most importantly, Husserl's "principle of principles" runs as follows:

> "No conceivable theory can make us err with respect to the *principle of principles*: that every originary presenting intuition (*Anschauung*) is a legitimizing source of that everything originarily (so to say in its 'personal [*leibhaften*] actuality') offered to us in 'intuition' (*Intuition*) is to be accepted simply as what it is presented as being, but also only within the limits in which it is presented there". (Husserl 1983, p. 44).

Thus, intuition (*Anschauung, der schaunde Blick, das schauende Ich*) is utilized by phenomenology in the broadest sense, as anything that manifests itself in consciousness in different *forms* of intuition, thereby making itself *evident* in various ways. *Schau*, seeing, especially "seeing the form" in Balthasar's texts could be rendered in Husserl's language as *Anschauung der Phänomene, Schau der Formen, der Gestaltungen*.

The concept of form can be understood as a version of the concept of phenomenon in phenomenology. If we look at the terminology of Balthasar, this is obvious: *Gestalt*, form, appearance, phenomenon go together; and even the more concrete definitions of form are consistent with the definitions of a phenomenon in the classical and related literature (Wertheimer 1925; Ellis 1938; Smith 1988). Balthasar describes the form as "as revelation of the depths" (Balthasar 1982, p. 118). The phenomenon, he continues, is an essential datum (in the sense of being ultimate and central); it has absolute reality; it possesses a defining quality; it shows, discloses, reveals itself, it remains at the same time hidden in its inner core; it is still obvious, overwhelming, and defining for the receiver of the form. Moreover, Balthasar repeatedly discusses the relationship between typical forms of religions (in the plural) and the unique form of Christianity, i.e., the form of Christ. The typical forms of religions are religious phenomena, as it turns out when Balthasar discusses Gerardus van der Leeuw's *Phenomenology of Religion* (Balthasar 1982, pp. 360, 496, 499). However, the form of Christ, which is the form of revelation, is *Gestalt* in an essential and unique sense. Its uniqueness cannot be subsumed under any typology. It is an ultimate phenomenon that is revealed by God and at the same time reveals God.

The concept of form has thus a threefold characteristic: First, it is a supernatural phenomenon rooted in the unique reality of God. Second, it is a form, i.e., a phenomenon, in the sense of an idea or essence. Third, this essential character is distinguished from its external and finite instantiation or expression, i.e., its "existence", while the two moments constitute a common whole. The original word *Gestalt* refers to this latter as it comprises, in the language of gestalt psychology, both the essential feature and its existential instantiation. *Gestalt* is both intellectual *and* sensual; it is essential *and* instantiated. It is precisely through instantiation that we have access to the essence. This understanding of *Gestalt* makes it possible to develop an aesthetical approach to theology, because it is the aesthetical realm where such an essential–existential unity prevails (Schindler 2004, pp. 163–255).

Here we find the key to the phenomenological understanding of *Herrlichkeit*. Glory is a *Gestalt* in the above sense, an instantiation of beauty that unites essence and existence. Of course, the translation of *Gestalt* as form may be misleading because *form* is originally the Aristotelian term for the essence or the idea. Yet it is precisely the Aristotelian understanding of the unity of form and matter (hylomorphism) that makes the conception of *Gestalt* even more pregnant. Hylomorphic unity is understood by Balthasar as the essential feature

of divine glory, *Herrlichkeit*, which is properly translated as splendor, something splendid, glorious, magnificent, and beautiful.

The central theoretical problem to solve here is the tension between the general or typical character of the hylomorphic unity and the uniqueness of the divine *Gestalt.* The problem is nevertheless not specifically theological. It belongs to the core of phenomenology to understand the unity and difference between essence and existence. Balthasar believes that phenomenology is an approach based on the rejection of real being. This is, however, a fatal misunderstanding of the work of all important phenomenologists beginning with Brentano through Scheler and Husserl up to Heidegger. As Husserl famously formulated, "The point is not to secure objectivity but to understand it". (Husserl 1970, p. 189). Phenomenology, already in the sense of Husserl, does not abolish reality, objectivity, or existence, but wants to understand them properly, i.e., not in the blind naivety of everyday life. Understanding aims at explaining the *meaning* of existence, which is related to its instantiation but cannot be collapsed into the latter. Collapsing meaning into what is meaningless is naturalism, historicism, or psychologism for Husserl (Husserl 1965); and phenomenology is the proper understanding and explanation of how meanings are constituted, including the meaning of the ultimate individual, i.e., existence. Such a meaning can never be conceived inductively; it is only *Anschauung*, intuition, that gives meaning in its immediate wholeness (Schindler 2009) that can be explained in philosophical prose.

Balthasar's problem is to find a way from the general typology of hylomorphic unity to the uniqueness of one single hylomorphic unity, i.e., the form of Christ or the form of revelation. His ultimate conclusion—which is only sporadically articulated in various parts of his work—is that the Christocentric uniqueness of form, i.e., divine *Herrlichkeit*, makes it possible and actual for there to be a plurality of created forms in the universe, a plurality described typologically but always with an emphasis on the existential uniqueness of instantiation. Balthasar applies descriptive phenomenology when he discusses, for instance, artistic or religious phenomena. He also applies the correlative approach of phenomenology when he acknowledges that the human mind is naturally capable of conceiving typologies and their instantiations, so much so that the mind and the form, essence and existence, prove to be correlative instances. Finally, Balthasar also practices a kind of transcendental phenomenology when he seeks to define the ultimate source of the universal *a priori* of correlation and its phenomenological typologies in and through—not the constitutive role of the transcendental *ego*, but—the ultimately unique, supernaturally unparalleled form of Christ as the center of divine revelation. The form of revelation is seen as the ultimate source of all typologies and their hylomorphic contents with a strong emphasis on the sensual instantiations of artistic beauty.

Let me note here that Balthasar repeatedly rejects the work, achievements, and claims of certain phenomenologists. This may be seen as an example of the "*hérésies husserliennes*" mentioned above; but it does correspond to the contents and implications of his endeavor. When Balthasar criticizes certain claims under the general label of phenomenology, it turns out that he is referring to a particular level or interpretation of phenomenology. When, for instance, he charges Husserl with "essentialism", he does not seem to realize that it is not the formally abstract concept that Husserl considers an essence. Moreover, there are generally formulated critiques in his works that target some versions of phenomenology without explicitly naming them. His approach seems to be that of an outsider who is confident of possessing an extra-phenomenological methodology. However, there is no systematically developed methodology in Balthasar's work (his method is "impressionistic", cf. Olson 2013, p. 596); he mixes historical description, rational analysis, philosophical evaluation, and dogmatic theology on a level that tends to confuse his readers. Behind this methodological confusion, as it were, there is still the pervasive influence of the phenomenological method expressed, most characteristically, in the notion of *Herrlichkeit*. Therefore, Balthasar's theological aesthetics can be seen as a unique kind of phenomenology that reinterprets both the phenomenological methodology and its evaluations in a significant way that cannot be properly understood without the phenomenological context of his time.

*4.2. History*

Balthasar engages the problem of history in the second series of *Theo-Drama* (see also Balthasar 1994). Again, the richness and complexity of the volumes of *Theo-Drama* surpass the limits of a simple analysis. In the first volume of *Theo-Drama* (Balthasar 1988), the author offers a typological description of various theological standpoints. The central concepts of his analysis—event, history, orthopraxy, dialogue, political theology, futurism, function, role, etc.—are types or "archetypes" (an expression he repeatedly uses) of a phenomenological approach. Gerardus van der Leeuw's similar though more detailed descriptions emerge as the background of this methodology (van der Leeuw 1986). This raises the problems that van der Leeuw sought to answer in the Epilegomena of his work. For Leeuw, Husserl's method involves "naming, systematic experience, *epoche*, clarification, and testimony" (Smart 1986, p. xi), a methodology Balthasar clearly, if implicitly, follows (including the importance of testimony, cf. Balthasar 1985, p. 267). What remains an open question for van der Leeuw—and thus for Balthasar—is the ultimate phenomenological source of such non-inductive concepts.

This descriptive-typological approach to the problem of history is continued in a different form in the subsequent chapters in *Theo-Drama I*. Significantly, Balthasar offers critical evaluations of individual authors, such as Kassner and Hegel. The reader gets the impression that Balthasar often agrees with these authors but tries to find points where his unique perspective can be brought into play. Here again we see the analytic-phenomenological approach which begins with concrete instances and aims to grasp an essential point, an archetype, which can be critically evaluated. The goal of such a critique is to understand the dramatic whole—in terms of an ultimate testimony—which is proleptically presupposed. This holistic approach is again deeply phenomenological and tacitly presupposes the higher levels of phenomenology mentioned above, i.e., correlative analysis and transcendentalism (in a theological perspective).

Balthasar's arguments are made very complicated by his zigzagging between the various aspects of phenomenology: typology, the study of paradigmatic cases (instantiations), correlative analysis, and the holistic approach. The latter is the most important in Balthasar's view as it concerns his critique of the literary genre of drama and its substitution with the real-theological drama of salvation. The difference between a dramatic role and a divine mission is the following: a dramatic role is a literary form and it remains within the framework of a finite representation of reality. Mission, however, is a theological category, rooted in the Trinitarian doctrine, that transcends finite roles in a drama and locates the actor in the real, historical drama of salvation. The archetype of mission is the mission of Christ; and by participating in this mission, one can realize one's mission, a vocation that is infinitely more significant than the dramatic role in a theater. Here again we see the holistic approach of phenomenology applied to the material of literary criticism with a theological complement.

The ultimate horizon of the dramatic theory of Balthasar is his analysis of the relationship between the immanent and the economic Trinity, i.e., the relationship between God's *ad intra* relations and his creative and salvific *ad extra* activity in the world. This is of course a most intricate theological puzzle. No wonder some of Balthasar's critics raise the problem of a theosophical insight here (Kilby 2012, p. 129), for it is indeed a central question how and to what extent we can know God's *ad intra* reality. Can we gain knowledge of these relations by a kind of induction beginning with the Christ of history? Or are we to offer an interpretation of ancient and contemporary authors in order to arrive at some knowledge of God's innermost reality? Or is it possible in faith to gain insight into this realm? (cf. Tóth 2024).

Balthasar outlined his answers in several contexts (e.g., Balthasar 1984, Introduction). However, if we follow an inductive methodology, our conclusions remain probabilistic. If we use literary criticism, we need logical analysis and also some kind of general logical matrix to come to a solid conclusion. Finally, insights given in faith are certainly possible, but then we postulate the knowledge of an ultimate horizon, which is the transcendental

realm in the phenomenological approach. Transcendental subjectivity—i.e., the focal point where our propositions become meaningful—cannot be swept aside even if we do not open the perspective of the ontology of absolute being. Here we can repeat Husserl's principle in a different form: the point is not to deny *transcendentality* but to understand it (cf. Husserl 1970, p. 189). If properly understood, it is possible to open transcendentality to the ontology of absolute being, or to a theological ontology, as Husserl already suggested (Husserl 2013, pp. 478–79). At the very least, we can say that there is a close parallelism between the holistic dimension of phenomenology and an analysis of the content of faith, a parallelism that requires more careful investigation than a rhetorical rejection.

The crucial problem with Balthasar's theo-dramatic perspective is this. He uses the ancient scheme of drama as a literary form to lead us to the drama of divine revelation, a drama that is no longer a literary form but the very structure of reality. But are we allowed to do such a *salto mortale*? Can we jump from the stage of literary history to the stage of the real history of salvation? It may be a misuse of the ancient literary form of drama (as a symbolic expression of reality as perceived by the ancients) for the purpose of theological explanation, in such a way that the former is sharply criticized and the latter is heartily accepted. Here the reader suspects the logical error of *metabasis eis allo genos*. Balthasar's reason for this *metabasis* is clear: his holistic understanding of history has always been a problem of the relationship between the *ad intra* and *ad extra* dimensions of God. This problem is even more sharply formulated as the form of revelation is intrinsically historical, i.e., rooted in the history of Jesus. To understand this history as the *ad extra* expression of *ad intra* processes and relations—Trinitarian relations—is again very close to the problem of historicity in phenomenology. This problem is never properly solved by Husserl and Heidegger (not even by their followers), but the former developed teleological phenomenology, a process phenomenology underpinning the wholeness of essences, and the latter offered the decisive importance of factual life, *Dasein*'s concrete reality with respect to event, *Ereignis*, in his later works (cf. Husserl 2013, pp. 478–79; Scheler 2017; Polt 2013a).

Obviously, Balthasar not only studied the authors of phenomenology but also borrowed their solutions. He even mentions that his approach is justified by the principle of the *spolia aegyptiorum* (cf. Exod 3:21–22; 12:35–36; O'Regan 2018, p. 54; cf. Betz 2023, p. xxiii). Reading St Augustine's passage on this idea (Augustinus 1834a, p. 63), one has to think that, for Balthasar, the unique discoveries of Heidegger, which have so profoundly influenced contemporary philosophy, were "unlawful" (*iniustis*). This idea might have had some plausibility if applied to some Hellenistic authors, but it tends to cause an intellectual and moral embarrassment in relation to a twentieth-century thinker (and even raises the question of plagiarism). What Balthasar wants, of course, is a proper theological interpretation of the findings of contemporary cultural figures (Balthasar 1982, Introduction). But where he most clearly uses Heidegger's thought is not in the theological but in the philosophical part (Balthasar 1985, see here the many uses of "unconcealment" [*Unverborgenheit*], one of Heidegger's central concepts). Balthasar might have seen his procedure as a reclaiming of what belongs to theology, since he accuses Heidegger of committing a reverse *spolia aegyptiorum*, i.e., of using theological material in his development of the concept of being (Balthasar 1991, pp. 429–50 [Heidegger the way back], especially 409). This criticism is both intellectually and historically problematic, because a more balanced evaluation would also require an analysis of the accusation of "popular Platonism" against Christian theology, as has been leveled—beyond Nietzsche's famous dictum—by a number of scholars from Franz Brentano to Jan Patočka to Endre von Ivánka ([1964] 1990) (see his *Plato Christianus*, published by Balthasar himself in 1964).

*4.3. Logic*

Balthasar's *Theo-Logic* is a complex work which, even more than the volumes of *Herrlichkeit*, reveals the author's eclectic style and, beyond that, his eclectic character as a thinker. Beginning with descriptive phenomenology, where it is the realm of mere phenom-

ena that stands in focus, the following can be said. Balthasar rejects that his methodology is eclectic and offers rather the label "integral" (Balthasar 2004a, pp. 15–19). What he means becomes clear as he addresses various features of a problem or issue and lets the reader gradually see the underlying unity behind the various partial descriptions. Balthasar is aware of the importance of essences, a central point in Husserl's phenomenology, but he always links essences to existences by pointing out the importance of integration. However, he tends to believe that essences are abstractions in phenomenology and not rather ideal types, or even more, archetypes or original phenomena. Had Balthasar realized that Husserl's *epoche*—to which he refers a few times (e.g., Balthasar 1998, vol. III, p. 115; Balthasar 1982, p. 499)—is not an act of abstraction but rather the effort to let things be seen as they are, i.e., not reducing them to some aspects, he would have almost certainly accepted Husserl's phenomenology in a more generous way (*pace* Schindler 2009). What he obviously accepts, however, is the typological analysis that he applies throughout his work. He offers a phenomenological analysis of love, bodily and spiritual shame, and he repeatedly points to the importance of phenomena in our understanding of reality. Even if embedded in the more general context of truth and being, Balthasar is obviously aware of the importance of the phenomenological method of revealing and analyzing essences.

Balthasar emphasizes the importance of facts as well. The ineffable nature of individual existence comes to the fore again and again as this is his way to show the unavoidable relation of essence to existence. While he considers existence in an essential way (as existence in the general sense), he emphasizes the presence of existence in all aspects of human life. Existence, he writes, is such a simple and elementary concept that it requires no explanation or analysis. However, Balthasar calls existence "a riddle" (Balthasar 1983, p. 499) and explains its nature as an opening to the "interiority of being" (Balthasar 2004c, p. 85). This is certainly a kind of phenomenological analysis, albeit a rather simplistic one. It is even more simplistic to say that for Husserl "philosophy can simply bracket existence from the outset and devote itself entirely to the investigation of its rightful field, namely, essence". (Balthasar 2004c, p. 105)

This brief critique of Husserl shows Balthasar's misunderstanding of the Austrian philosopher. Husserl's "bracketing" existence is not, first of all, an ignoring of factual experience. Rather, it is a recognition of the importance of such experience as that which is to be understood (not "to secure", *sichern*, Husserl 1970, p. 189). Husserl offers the method of the *epoche* and the reductions not to get rid of facts but rather to understand them *as* facts. At the same time, factual experience for Husserl is always put into the context of the intentional correlation of the *noema* and *noesis*. For in the everyday attitude, we are given facts always in the context of a certain attitude, *Einstellung*; the philosophical, i.e., holistic, attitude to facts aims at their wholes in which factuality is one segment. What we bracket is the aspectual dimension of a fact so that we can, as it were, liberate its full reality in what we perceive as a phenomenon.

Balthasar continues his interpretation of phenomenology with clear examples of correlative analysis. Existence is always conceived in relation to essence and *vice versa*. This is certainly very close to what Husserl and other phenomenologists say with the important addition that insofar as we understand this correlation, we are in the context of intentional analysis. That is, for Husserl, phenomenology does *not* get rid of concrete existence. On the contrary, phenomenology is the way to understand it. Without this understanding, a concrete existence is not existence at all. And what we understand, the essence, is never an abstraction (to repeat this point) but rather a whole of which real existence is only one dimension. It is always on the basis of this understanding that explanation can be given in such a way that the essence emerges as the very content of this concrete explanation. When Balthasar discusses existence and essence, among other things, he is dealing with concepts, notions, which are better understood as phenomena, i.e., primordial wholes belonging to the phenomenological context of correlation.

As mentioned, Robert Givord, the French translator of Balthasar's *Wahrheit*, rightly understood that the work (which later became the first volume of Balthasar's *Theo-Logic*) was

indeed through and through phenomenological. The tenor of Balthasar's understanding of truth is his interpretation of Heidegger's notion of *aletheia*, unconcealment. Unconcealment in Heidegger is not only the genuine nature of truth but more importantly the nature of a phenomenon (Heidegger 1996, sct. 7). A phenomenon is that which shows itself in and through itself. Phenomenology is the analysis of this self-showing in the phenomenological circle, which begins with the natural fact and arrives at the fully understood phenomenon. Balthasar is so much under the influence of this phenomenological paradigm that one could in fact find it problematic that he does not even mention the name of Heidegger in these passages. Even if he interprets in a certain way this fundamental Heideggerian insight, what he offers is still a kind of phenomenology, in this case already a correlative analysis. This analysis is carefully developed in the volumes of *Theo-Logic* so that truth turns out to be the truth of God and ultimately the Spirit of Truth. However, the philosophical foundation of his theological elaborations is simply phenomenological.

For Heidegger, the ultimate horizon of truth is being as such. In his middle period, being turns out to be event, *Ereignis*, and in his later period being is already the mysterious *Seyn*, be-ing (Polt 2013b, p. 148). In all these developments, Heidegger reaches a unique philosophical vision which goes far beyond any elementary existentialism or everyday realism. This vision has an important theological aspect called ontotheology (Thomä 2013, p. 260). Balthasar reflects on the problem of ontotheology a few times (Balthasar 2004a, p. 92; 2004c, pp. 134–35, fn 10). He seems to view the charge merely in terms of the projection of finite beings onto the infinite divine Being. However, Heidegger's argument goes much deeper in that he claims that the entire traditional concept of being (finite and infinite) is misguided. In this way, Heidegger continues the critique already contained in Husserl's theory of *epoche* and the reductions, a theory that aims at a proper understanding of reality rather than reducing it to some of its aspects. Balthasar's approach appears to fit in with this understanding, but the theological whole that he maintains does not seem to answer the radical challenge of the problem of ontotheology—a problem that needs to be addressed in a separate paper as it is more serious than superficial rejections and misinterpretations might suggest.

### 4.4. Love

Balthasar's analysis of love is one of the best examples of his implicit application of phenomenology. First, the analysis of love in *Love Alone is Credible* (Balthasar 2004b) shows a descriptive approach to the phenomenon of love. Second, Balthasar offers a typology of various forms of love. Third, he proceeds to a transcendental interpretation of love, which includes a paradoxically transcendent conclusion. This conclusion leads back to the preliminary analysis where Balthasar explains the natural form of love in terms of the overwhelming influence of divine love.

Balthasar emphasizes that love in its natural form is only fragmentarily present in human persons. It is only divine love that fulfills the broken nature of human love. Without God's absolute love human persons cannot attain a more consistent form of love and cannot fully love that which is "wholly other" (Balthasar 2004b, pp. 11, 72, 75, 140). The human capacity of love is interpreted as an implication of divine grace, which "necessarily includes in itself its own conditions of recognizability and therefore brings this possibility with it and communicates it" to human persons (Balthasar 2004b, p. 75). However, Balthasar does not seem to recognize that a loving relationship between persons necessarily involves the original and genuine capacity for love that is intrinsic to the persons who love each other. If love is only an epiphenomenon, if it is not intrinsic to a person, he or she cannot truly love another person. Moreover, in order to attain the full form of love (i.e., love in its true nature), such persons must be free; the love of one who is forced to love is invalid. That is to say, even though we emphasize the ultimate theological framework of love, love must be natural to human persons and must also be freely realized in relation to other persons, including the persons of God.

Of course, we can still maintain that divine love "necessarily includes in itself its own conditions" when it reaches out to human persons. But then we must see clearly that if we overemphasize the dependence of finite love on infinite love, if we undermine the naturalness and freedom of human love, we also undermine the validity of the divine framework within which true love takes place. If the finite is called to love the infinite, then the finite must have a true independence. A one-sided emphasis on the infinite abolishes the reality of the finite; and a finite deprived of its natural independence loses its intrinsic capability to be open to the infinite. While it is true that even nature is grace (in a sense), nature must have its freedom and inclination to love that which is beyond nature, i.e., the infinite. In other words, the finite does not exclude the infinite; nor is the transcendental opposed to the transcendent. Rather, the transcendental and the finite open up to the transcendent and the infinite like a bright window to the horizon of the ocean.

Balthasar offers a typology of love. One by one he analyzes love as divine revelation; love as justification and faith; love as deed; love as form; and love as the light of the world. These subjects (each being the title of a chapter in the book mentioned) contain descriptions of various concepts of love, of which especially the concept of form appears to be eminently important. This form is a *Gestalt* (here the German text uses also *Form*) with an inner unity that cannot be divided into parts; that is to say, the historical forms of love, the subjective experience of love, and even divine love belong to the same unity of form. This form has a paradigmatic expression in the historical figure of Jesus and an absolute unity in the inner loving relationships of the Holy Trinity. In other words, this transcendent level of love is at the same time transcendental, i.e., it permeates the nature of human beings. This capacity, nevertheless, can be properly realized if and only if divine love activates it.

Balthasar first distinguishes two historical patterns of love which he judges to be partial and therefore also unsuccessful. The first is termed the cosmological reduction in which the capacity of love is an intrinsic feature of the universe. Divine love is paradigmatically present in the universe and can be perceived in its fullness along the lines of this cosmic pattern. For both Plato and Aristotle, the universe is governed by divine love, so love is seen as a natural extension of the cosmological structure of the universe. However, Balthasar explains, divine love cannot be described in terms of the world. To do so would be to reduce love to nature, which is unacceptable. The second historical pattern is anthropological love. This love cannot express genuine love either, for then human love would be the measure of divine love, which must again be rejected. It is only divine love where love is fully realized and expressed, concretely in the life, death, and resurrection of Christ. Christ is absolute love both in terms of history and as the expression of the perfect love of the second Person of the Trinity. What we see in this division—love as cosmological "reduction", anthropological "reduction", and finally love in its absolute form in divine revelation—is indeed a phenomenological classification with an emphasis on the absolute fullness of love that is necessarily expressed in real history in one concrete person:

> "God, who is for us the Wholly Other, appears only in the place of the other, in the 'sacrament of our brother'. And it is only *because* he is the Wholly Other (in relation to the world) that he is at the same time the *Non-Other* (Cusanus: *De Non-aliud*), the one who, in his otherness, transcends even the inner-worldly opposition between this and that being. Only because he is *over* the world is he *in* it. But being *over* it does not deprive him of the right, the power, and the Word, to reveal himself to us as eternal love, to give himself to us and to make himself comprehensible even in his incomprehensibility." (Balthasar 2004b, p. 150)

Balthasar, thus, completes his phenomenological analysis with a statement that implies the simultaneously transcendent and transcendental nature of love. For if God is in the world because He is over the world, then nature, including transcendentality, is given its proper place. In this sense, neither nature nor transcendentality, not to mention the transcendental *ego* is opposed to God. That is to say: phenomenology is not against theology; rather, it helps understand theology better. Although Balthasar seems to sweep this conclusion aside in several other places in his work, this little book on love is the

clearest example of how he understands and applies classical phenomenology. And his phenomenology consists in this case is not only in the use of the insights of Husserl and Scheler, but even more so in the use of the spirit of phenomenology with an important addition: the transcendental or *a priori* reality of love as revelation is actually the disclosure of divine love in such a way that in this disclosure we recognize the original source of love in God.

It is within this framework, then, that we must understand Balthasar's initial reference to Scheler's idea of "the pure self-giving of the object" (Balthasar 2004b, p. 12) as a pattern of his own vision of love. However, it is a philosophical error to think that any pure self-giving of an object eliminates the conditions of possibility of self-giving—even though these conditions are taken from the source of the self-giving object. These are two separate questions: one is about the origin of a condition, and the other is about the fact and meaning of that condition. Phenomenology, even in Scheler's version, emphasizes pure self-giving not against pure receptivity, but rather as a correlate to that receptivity. Scheler developed this point into the concept of the "eternal in man", an idea that is criticized by Balthasar in the above work (Balthasar 2004b, p. 98). But even in his critique, Balthasar seems to follow Scheler's methodology in his own way (with his characteristic misunderstanding of the "phenomenological bracketing", cf. ibid., p. 12).

Finally, let me note the following: Balthasar presupposes a *concept* of love, the subject matter of his investigations, i.e., a concept that is intrinsically meaningful, *sinnhaft*, and treated appropriately in its meaningful nature, i.e., in terms of a phenomenology of meaning. The ultimate source of this meaning is the Wholly Other, which, as it turns out, is necessarily Non-Other, i.e., it pervades nature and man alike, it is present in nature, in human beings, in history, in reason and in faith. In such an approach, Balthasar presupposes and applies phenomenology, even though with some idiosyncratic accents. Balthasar's phenomenology of love still reads a little like Schiller's great poem *An die Freude* in which *Freude*, divine joy, permeates and animates the entire universe, yet it is at the same time transcendent to it: "You are embraced, humankind! / Kissed you are in all the world! / High above the skies, my friends / Your Father must live and thrive!" (My translation).

## 5. Supernatural Phenomenology

We can describe Balthasar's understanding of phenomenology with his own expression: what he offers implicitly and explicitly is "supernatural phenomenology" (Balthasar 1954, p. xvi; for an interesting application, see Moss 2004). I use this title in the sense of the different levels examined above: as a descriptive analysis, as an analysis based on correlation, and as a transcendental method, which are articulated in different ways by the most important phenomenological authors. Balthasar's theology is clearly a kind of phenomenology in the sense of all these levels. But instead of transcendental phenomenology, he offers "supernatural" phenomenology. How can we understand this claim? What is its philosophical significance? What are the consequences for Balthasar's theological system? As mentioned above, I will not discuss here the narrowly theological aspects of Balthasar's work, but I will try to outline answers to the other questions, and partially even to the last one.

Considering the logic of phenomenology in its classical period, it is obvious that it was never necessarily atheistic as Sartre and Janicaud among others claimed (Sartre 1956; Janicaud 2015, chp. 12). It is more plausible to say that phenomenology *ab ovo* tended to become a philosophical theology. The development of the work of some of the classical phenomenologists, especially Scheler's philosophy, clearly shows this and Balthasar also confirms it (Balthasar 1998, vol. III, pp. 126–52). Outside observers, for instance Etienne Gilson, also noted this tendency (Gilson 1987, pp. 365–77). We can even say that phenomenology was open to a new approach to divine revelation, as Maurice Merleau-Ponty suggested (Merleau-Ponty 1970, p. 46) and as is evident in contemporary developments (Graves 2021; Marion 2021; Mezei 2021). Balthasar argues against the separation of categorical and transcendental revelation (cf. Mezei et al. 2015, pp. 117–18),

i.e., Christian revelation in the dogmatic sense and general or natural revelation. Of course, revelation is in itself one, but in this unity, there are layers, levels, and aspects that can be clearly distinguished so that their unity may become *even more* evident. Both the unity and the manifold character of the concept of revelation can be conceived by appropriate phenomenological procedures.

Analytic phenomenology describes facts and their relations in terms of concepts and their relations. There is no other way in any theoretical procedure: we deal with concepts, terms, and their relations in every theory. The question is whether these concepts and their relations are merely given as results of inductions. Phenomenology denies this and points to the inescapable formal dimension of our concepts, i.e., their essential nature. The result is, at an elementary level, a periodic table of types or archetypes. In the proper understanding of a type, everything depends on how we define this term. If a type is not inductive, it can be either *a priori* given or intuitively perceived. Good phenomenological authors reject the formal *a priori* dimension. What remains is the intuitive perception of essences in and through their existential instances. Such a procedure nevertheless presupposes a certain matrix of essences, phenomena, and their correlations. We participate in this universal correlation in which essences are intuited and thus constitute the network of being and knowledge centered on the transcendental pole of identity.

Balthasar applies analytic phenomenology throughout his works. Where he departs from good phenomenology is in the ultimate source of the archetypes and their correlation. Up to this point, however, the entire methodology of integration, his multidimensional approach in which various subjects are elevated in the transcendent center, is followed in the characteristic Balthasarian style. This style combines all kinds of analysis—literary history, philosophy, theology, drama, music, and poetry—in order to disclose the structure of divine revelation in its fullness, i.e., both in its universal presence and in its historical uniqueness. The latter is at the heart of Balthasar's views, i.e., a factuality and historicity that can also be found massively developed in some phenomenologists, such as Heidegger or Ricœur. In this multidimensional approach, Balthasar repeatedly combines analytic phenomenology with correlative analysis and correlative analysis with transcendent conclusions in order to underpin his kind of dogmatic theology. In these arguments and elaborations, As a theological mind, Balthasar occasionally touches on important philosophical insights, such as the ultimate form of truth in personal confession. (Balthasar 1985, p. 267 [Confession]).

In particular, Balthasar's supernatural phenomenology is a phenomenology of the personality of the saints (Moss 2004). In the lives and works of these persons, we not only see the factual aspects of sainthood but also the transcendent source of sainthood in the person of Christ and, ultimately, in the holy community of the Trinity. This phenomenology is supernatural because all the phenomena it considers, analyzes, and explains are placed in the context of the person of Christ and the works of the Holy Spirit. The ultimate focus of this phenomenology is the transcendent fact of God's absolute uniqueness. This is correlated with the uniqueness of the particular saint in that not only is God the agent of sainthood, but the saint also contributes to God's saving work. Between the person of the saint and God there is an intimacy or intertwining (*Ineinandersein*) that instantiates the intimacy of the persons of the Trinity in a historically concrete situation. In this way, history in its real occurrence is inseparable from the *ad intra* relations of the Trinity. The essence of Christianity is precisely to produce such historically unique personalities with their community in the Church and to connect and reconnect this community with the Holy Community of the Trinity.

What is the philosophical significance of Balthasar's supernatural phenomenology? It is a full-fledged phenomenology within the context of a Trinitarian theology in which various aspects of a philosophical phenomenology can find their place. Balthasar's focus is on divine revelation, and thus his supernatural phenomenology is a phenomenology of revelation. The inherent tendency of the phenomenological movement to become a philosophy of revelation is clearly articulated in Balthasar's use of the phenomenological method. Since he himself claims that "without philosophy, there can be no theology",

(Balthasar 1985, p. 7), his supernatural phenomenology can certainly aspire to the role (or mission) of being such a philosophy. Nevertheless, this phenomenology is profoundly theological and thus reveals the intrinsic theological turn of phenomenology.

Since the focus of the present discussion is not primarily theological, suffice it to say that in various theological evaluations of Balthasar's thought, the important philosophical dimension is often neglected. Of course, one can see Balthasar's ongoing, implicit and explicit argument with and against Hegel and Heidegger (to mention only these authors), but there is a tendency in the secondary literature to exaggerate Balthasar's critique. Exaggeration here means, first, that it is often forgotten that the best form of reception is critical reception; second, that Balthasar is often wrong in his evaluation of philosophers; third, that even if he is right, this does not mean that the overall picture we have of a particular philosopher in Balthasar's works is the only reasonable interpretation. Neither a theologian nor a philosopher is without mistakes, missteps, or even more serious errors. As Goethe wrote, "Man errs as long as he strives" (Goethe 1963, p. 87). This must be emphasized here not only because Balthasar loved Goethe so much that he quotes him more often than any philosopher (cf. Balthasar 1998, pp. 407–515; Balthasar 1991, pp. 339–408), but also because Balthasar himself is so overwhelmingly significant as a thinker. His supernatural phenomenology is a unique expression of the inner drive of the phenomenological mindset to reach the ultimate source and fulfillment of all phenomena.

An alternative label for Balthasar's supernatural phenomenology would be "sacramental phenomenology" (Rivera 2022, pp. 8–9). If we accept the latter term, it seems to be possible to emphasize that what is supernatural is immanently expressed in what is sacramental. The phenomenology of the sacramental entails the traditional levels of phenomenology—the analytical, the correlational, and the transcendental—where the emphasis is not on the supernatural, understood perhaps in terms of ontotheology, but rather in terms of the self-disclosure of absolute reality. Sacramental phenomenology, then, is a phenomenology that leads to the absolute so that the absolute proleptically guides the human mind to conceive and understand the integral wholeness of reality without, nevertheless, claiming to comprehend it fully. It seems that Balthasar had a similar idea in mind when he spoke of a "phenomenological reading" of Maximus Confessor, a reading that reveals "the inner cohesion of the divine plan of salvation, read phenomenologically from the facts of sacred history". (Balthasar 2003, p. 222; a similar conception is described in Balthasar 1954). "Reading from the facts of history" is very similar to reading from phenomena in order to understand the nature of reality in its entirety, in which the reading mind is also organically involved.

## 6. Apocalyptic Phenomenology

Balthasar's famous first work, *Apokalypse der deutschen Seele*, uses the term "*Apokalypse*" in the fundamental sense that, according to the authors he discusses, a certain level of thought, art, meditation, or even action represents an *Endform*, the ultimate form of all possibilities, a form in which the origin and end of reality are revealed. Balthasar's sympathetic reading of some major authors of German literary and philosophical history shows that he took this claim seriously. At the same time, he criticizes the extreme features of these thinkers on the basis of a theological eschatology that excludes secular millenarianism. Like Hans Jonas or Paul Ricoeur (Jonas 1966, p. 254; Ricœur 1980, p. 139), Balthasar also applies the label "Gnostic" to various millenarian aspirations in literature and philosophy. Even in phenomenology, he discovered the urge to reach the *Endform* of thinking, after which history—literature as well as philosophy—ends (Balthasar 2004b, pp. 69, 105; cf. O'Regan 2001, p. 14).

It is interesting to note that the third volume of Balthasar's *Apokalypse* was published in 1939, the very year when the truly apocalyptic events of world history broke out. Balthasar's reluctance to accept the prophecies of the *Endform* philosophers was met with a powerful counterblow from historical existence. In the global collapse of order and the fury of war, the thinkers and writers of the *Apokalypse* seemed to be confirmed in more ways than one,

as if the thinkers and poets that Balthasar explored were signaling the coming collapse in which humanity also reaches its *Endform*. It is therefore legitimate to use the expression "apocalyptic phenomenology" as referring to the understanding of the phenomenological *Endform* that so deeply influenced Balthasar's thought. In my understanding, however, apocalyptic phenomenology is not about a particular historical situation; not even about an erroneous form of human thought. Rather, it is the understanding that in phenomenology the historic development of philosophy reaches a level where reality is being revealed with unprecedented depth and power. This is not true for every phenomenologist and with the same force. But the inner structure of phenomenology indicates in various ways the rise of a truly philosophical *Endform*. Apocalyptic phenomenology—indeed, a continuation of Balthasar's underlying insight—is one of these forms.

More concretely, apocalyptic phenomenology in the sense I understand this term seems capable of demonstrating two things: first, that phenomenology is at its core theologically oriented in a particular way, and second, that Balthasar's phenomenological understructure could be construed in a more consistent way. As to the first, apocalyptic phenomenology is the understanding and explanation of the fact that the center of phenomenology has always been divine revelation in the philosophical sense (Graves 2021; Mezei 2021; Rivera and O'Leary 2024). It is the self-disclosing nature of reality that apocalyptic phenomenology conceives and discusses, a self-disclosure only partially realized in nature or mind. The unrestricted fullness of self-disclosure occurs in its *sui generis* realm, i.e., in revelation *per se*. When we think phenomenologically, it is this *sui generis* realm that is approached through the various levels of the phenomenological method considered above: the analytical, the correlational, and the transcendental.

As to the second point, the transcendental aspect of phenomenology cannot be by-passed if we are to arrive at understanding; and understanding is always given in terms of concepts, notions, and ideas as *Gestalt*-like wholes comprising their various aspects and dimensions. The main problem with the transcendental *ego* is not that it can be eliminated—it cannot—but the mistaken view that the transcendental *ego* as the ultimate moment of self-disclosure does not open up a more overarching level of reality to which the *ego* is organically connected (as the seed is connected to the full-grown plant, cf. 1Cor 15:37 ff.).

The ontological turn in the history of phenomenology emphasized that the transcendental realm is the revelation of the ontological, i.e., the sphere of the meaning of being. The turn to alterity in phenomenology meant to call attention to the fact that this ontological realm is not *ego*-centered but contains otherness, *l'autre* or *l'autrui*. In the theological turn, the ultimate horizon of phenomenology is seen as the implication of saturated (better: supersaturated) phenomena that cannot be contained in intentional acts. Revelation is identified as that which explains the supersaturation of phenomena, and divine revelation is conceived as the most probable explanation of the source and content of saturation.

However, the supersaturated notion of revelation is inductive and thus only proba-bilistic. It cannot escape the fundamental problem Heidegger identified in *Phenomenology and Theology* (as mentioned above, see Heidegger 1998, pp. 39–63), which can be translated into the terms of ontotheology (i.e., the thesis that the entire traditional notion of being is misguided and its application to God results in an idolatrous concept). In this way, Heidegger continues the critique already contained in Husserl's theory of the *epoche* and the reductions, a theory that aims at a proper understanding of reality rather than reducing it to some of its aspects. Even if we develop the supersaturated notion of revelation at the level of the highest phenomenon and ascribe to it a phenomenality *par excellence*, we are still trapped in the realm of induction and thus also in a secondary, ontotheological approach to revelation. But revelation cannot be secondary, not even first in the sense of a member *par excellence* of a series or a set. Revelation is *sui generis* and it can only reveal itself as such. It cannot be *autrement qu'être* or coming *d'ailleurs* because it is *non aliud* and cannot be departmentalized into otherness and sameness (pace Lévinas 1974; Marion 2021; and see Balthasar 2004b, p. 122 [Recapitulation]). Of course, divine revelation can be described as a gift. But it is more like the gift of an engagement ring. An engagement ring is certainly a

gift, but its offering is naturally prepared and opens up the perspective of a life together, with its inexpressible intimacies and fruitfulness. As a gift, it can be freely accepted or rejected, and freedom is the most important and ultimately decisive natural condition for any gift, natural or supernatural.

Three different points can be made here regarding the importance of apocalyptic phenomenology. First, phenomenology is not at all a theory of induction. It is a theory of immediacy (it is not about an *il y a*, but about a *da*). Second, by its very definition, divine revelation cannot be the proper object of conjecture. Revelation by its very nature is always immediate even if we cannot perceive this immediacy under any circumstances and in its full meaning (cf. "The Ontological Argument for Revelation", Mezei 2017, pp. 30–32). Third, an inductive approach to the full fact of revelation cannot find its object, which is to say that no symbolic, hermeneutic, or mediated approach to revelation can arrive at what is properly called revelation. In other words, only revelation can reveal what revelation is. It is relatively easy to argue for the first point by means of Husserl's "principle of principles" (Husserl 1983, p. 44), which has never been overwritten by Heidegger's "hermeneutics" or "existentialism", since the latter sought a broader understanding of what is directly (t)here (*da*). On the second point, Balthasar's argument about the integral or undivided character of divine revelation is illuminating. We may add that the very possibility of revelation implies its actuality, even if we cannot understand the actuality of revelation in the full sense. As to the third point, the proper explication of the logic of induction may be a sufficient argument for that point, but let me add the following: no form of mediation can lead to immediate perception without the proleptic *a priori* givenness of the content of immediacy.

Apocalyptic phenomenology can be called the culmination of the phenomenological movement (Mezei 2022a) in that in it the core of reality is disclosed. Insofar as Balthasar outlined a historical form of apocalyptic thought, his approach can be used in the proper explanation of apocalyptic phenomenology. If I add that the core of reality that apocalyptic phenomenology is to unveil is *nouveauté novatrice* or renewing newness (an expression introduced by Miklos Vetö), I think that an important point about this kind of phenomenology is fixed (Vetö 2018, p. 32). Even more important, however, is that the essential expression of this newness is our self-renewal, a renewal "in the spirit of our mind", realized by the "new man" (Eph 4:23–24), i.e., the follower of apocalyptic phenomenology.

## 7. Conclusions

In this essay, my goal has been to explore the impact of phenomenology in the work of Hans Urs von Balthasar. It is obvious that he has used the phenomenological method repeatedly and importantly in his theology and philosophy. His most important theological aesthetics is a phenomenological endeavor, while he reinterprets the phenomenological method in essential points, especially in that he does not follow Husserl's way of understanding existence. At the same time, Balthasar makes massive use of Heidegger's phenomenological ontology, so much so that even Heidegger's terminology appears at crucial junctures in his work. In general, Heidegger's explicit influence is decisive for Balthasar, although he tends to ignore this fact. In his critique of phenomenology, however, Balthasar shows that the cosmological and anthropological reductions he criticizes still shape his outlook, especially the externalist and objectivist narrative of his theology—a consequence of his misunderstanding of Husserl's *epoche*. Finally, we have seen how a more consistent phenomenology, i.e., apocalyptic phenomenology, can fulfill the intrinsic requirement of the phenomenological and methodological aspects that Balthasar used in his theological works. Apocalyptic phenomenology follows not only the first sentence of Augustine's dictum, a favorite passage for Husserl, but also the second, which shows the way from interiority to true absoluteness:

*"Noli foras ire, in teipsum redi; in interiore homine habitat veritas. Et si tuam naturam mutabilem inveneris, transcende et teipsum"*.—"Do not go out, but return to yourself, for it is in the inner man that truth dwells. And if you find your nature mutable, transcend yourself!" (Augustinus 1834b, p. 39).

However, interiority and transcendence form a common whole, an *Ineinandersein*, which cannot be properly conceived or described by a one-sided monism or a denaturalized dualism.

**Funding:** This research received no external funding.

**Data Availability Statement:** No new data were created or analyzed in this study. Data sharing is not applicable to this article.

**Conflicts of Interest:** The author declares no conflicts of interest.

## Note

[1] I use the short form of Hans Urs von Balthasar's name as Balthasar. This is not intended to be derogatory in any way. The German nobiliary particle "*von*" is repeatedly omitted in texts referring to personalities of noble patrilineality. For example, we rarely say "von Humboldt" instead of "Humboldt", "von Kleist" instead of "Kleist", "von Dohnányi" instead of "Dohnányi". The regular dictionary form of these names is listed under the first letter of the family name.

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
