# Peer review of "Phénoménologie de la Vérité: The Phenomenological Roots of Hans Urs von Balthasar’s Theology"

_religions, doi:10.3390/rel15040409_

Round 1

Reviewer 1 Report

Comments and Suggestions for Authors

This article is very good and deserves to be published. It is original and well-argued, with thorough documentation. It demonstrates how Balthasar develops a phenomenology of the surnaturel in dialogue with the phenomenological currents of his time. The article illustrates how this phenomenology does not allow for the separation of categorical revelation from transcendental revelation. Therefore, it is a phenomenology of revelation that delves into the phenomenology of the lives of saints, a phenomenology of sacraments, and a methodology that the author connects, in his terms, to an apocalyptic phenomenology. I just suggest a few things regarding the form and content to improve it.

Regarding the form:

p. 8 – “Finallly” has only two ‘l’s.

p. 15 – “as Heidegger or Paul Ricoeur” - the author should choose to either use only the last names of both authors or the first and last names of both.

Regarding the content:

The author states on page 15: “this phenomenology is profoundly theological and thus reveals the intrinsic theological turn of phenomenology.” In my view, Balthasar’s phenomenology consists more of a phenomenological turn within theology than a theological turn within phenomenology. Indeed, as the author acknowledges, Balthasar is concerned with Revelation – his question is essentially theological. Rather than demonstrating that phenomenology is achieved by a theological turn (although Janicaud’s expression is a critique of this turn), it is about approaching Revelation from a phenomenological framework.

Moreover, at the beginning of the article, the author mentions that Balthasar read and appropriated classical phenomenology. While accurate, I believe it is crucial to emphasize Ricoeur’s expression in À l’école de la phénoménologie: ‘Husserlian heresies.’ Phenomenology follows Husserl and his heresies. This term is more fitting to describe how Balthasar reads the phenomenologists of his time.

I also suggest removing the name ‘Michel Henry’ on page 2. Michel Henry had limited knowledge of Balthasar, which is irrelevant to his work, despite Marion’s desire for him to read Balthasar. The names Marion and Falque are appropriate.

Finally, when the author describes phenomenology by stating: “phenomenology does not get rid of concrete existence. On the contrary, phenomenology is the way to understand it” (p. 12), it would be beneficial to distinguish between understanding and explanation. To refine the sentence “This phenomenology is supernatural because all the phenomena it considers, analyzes, and explains are placed in the context of the person of Christ and the works of the Holy Spirit. – explains?” (p. 15), it would be good to establish a difference between understanding and explaining.

Author Response

Answer to reviewer 1

Thank you for your important comments. I have corrected the minor typographical errors you point out.

You write: "In my view, Balthasar's phenomenology consists of a phenomenological turn within theology rather than a theological turn within phenomenology."

As I explained in my article, theology used phenomenology much earlier, e.g., by Tillich, Bultmann, and-even before Balthasar-Rahner.

You write: "I think it is crucial to emphasize Ricoeur's expression in À l'école de la phénoménologie: 'Husserlian heresies'. Phenomenology follows Husserl and his heresies. This term is more apt to describe how Balthasar reads the phenomenologists of his time."

You are absolutely right. I have added references to des hérésies husserliennes 2 times in the article (and Ricœur's work in the bibliography).

I removed M. Henry's name from 2 paragraphs.

I inserted the term "explanation" 4 times in order to satisfy your perfectly legitimate point.

Reviewer 2 Report

Comments and Suggestions for Authors

Line 216 missing possessive ("Balthasar's" not "Balthasar")

Line 233 Henry seems oddly placed with the hermeneutical philosophers, and might be better situated with the second group of phenomenologists, insofar as he is often quite critical of hermeneutics, which he understands as almost universally guilty of "ontological monism." Additionally, his own philosophy emphasizes the uninterpreted (unmediated) direct experience of self-affectivity. 

Line 256 -257 this line proposes the "life, death, and resurrection of Jesus" as the "immediate basis of theology." It might be more precise to mark this as the "immediate basis of Christian theology" as it undoubtedly does not form the revelatory basis of other theological traditions.  

Line 597 Here "ontotheology" is evoked but not defined. It might be helpful to briefly define the term for unfamiliar readers at this point. 

Lines 875-948 The introduction of apocalyptic theology as an alternative to immanentism and transcendentalism functions fairly enough, but the discussion seems to veer away from von Balthasar. It is unclear to the reader why this approach particularly coheres with or extends von Balthasar's project specifically. A clearer link to von Balthasar here would be helpful. 

Line 952 "Hans Urs von Balthasar" not "Hans Urs Balthasar"

Throughout: It is usually good practice to retain the particle "von" in Germanic surnames (i.e. "von Balthasar" rather than "Balthasar"). This is complicated by the fact that "Balthasar" alone has become the norm in some secondary English language scholarship on von Balthasar, but this fact annoys me. Take or leave this advice. 

Author Response

Answer to reviewer 2

Thank you for your important comments. I have corrected the minor typological errors you pointed out. Michel Henry originally belonged to structuralism and Marxism, only later did he begin to move toward theologically inclined phenomenologies. So I do not count him as a hermeneutic philosopher.

Theology is certainly Christian where I discuss it in my article.

The term ontotheology occurs 6 times in my article, and I briefly explain its meaning on p. 17 (manuscript page numbers), but now I have also added a brief insert about its meaning after its mention on p. 24.

The meaning of apocalyptic phenomenology and its connection to Balthasar's phenomenological thought is explained in the original version of my article. However, to make this connection clearer, I have deleted two sections (relating to pantheistic and overtranscendent interpretation) and added a new section on Balthasar's understanding of the Apokalypse as Endform. I now initially define apocalyptic phenomenology as a reformed version of Balthasar's Apokalypse as Endform.

I have inserted a footnote at the first use of Balthasar's name, explaining the omission of "of." Here is the text:

"I use the short form of Hans Urs von Balthasar's name as Balthasar. This is in no way meant to be derogatory. The German nobiliary particle 'von' is often omitted in texts referring to personalities of noble patrilineality. For example, we rarely say "von Humboldt" instead of "Humboldt," "von Kleist" instead of "Kleist," " von Dohnányi" instead of "Dohnányi”. The regular dictionary form of these names is not listed under "v", but under the first letter of the family name.